# Effects of Spraying KH_2_PO_4_ on Flag Leaf Physiological Characteristics and Grain Yield and Quality under Heat Stress during the Filling Period in Winter Wheat

**DOI:** 10.3390/plants12091801

**Published:** 2023-04-27

**Authors:** Jinpeng Li, Zhongwei Li, Xinyue Li, Xiuqiao Tang, Huilian Liu, Jincai Li, Youhong Song

**Affiliations:** School of Agronomy, Anhui Agricultural University, Hefei 230036, China; jinpeng0103@ahau.edu.cn (J.L.);

**Keywords:** dry matter, grain yield and quality, heat stress, spraying KH_2_PO_4_, winter wheat

## Abstract

As one of the most important wheat-producing areas in China, wheat is prone to heat stress during the grain filling period in the Huang-Huai-Hai Plain (3HP), which lowers yields and degrades the grain quality of wheat. To assess the effects of spraying potassium dihydrogen phosphate (KH_2_PO_4_) on the physiological traits in flag leaves and grain yield (GY) and quality under heat stress during the filling period, we conducted a two-year field experiment in the winter wheat growing seasons of 2020–2022. In this study, spraying water combined with heat stress (HT), 0.3% KH_2_PO_4_ (KDP), and 0.3% KH_2_PO_4_ combined with heat stress (PHT) were designed, and spraying water alone was used as a control (CK). The dates for the spraying were the third and eleventh day after anthesis, and a plastic film shed was used to impose heat stress on the wheat plants during the grain filling period. The results showed that spraying KH_2_PO_4_ significantly improved the chlorophyll content and net photosynthesis rate (Pn) in flag leaves compared with the non-sprayed treatments. Compared with CK, the Pn in HT decreased by 8.97% after heat stress, while Pn in PHT decreased by 7.44% compared to that of KDP. The activities of superoxide dismutase, catalase, and peroxidase in flag leaves were significantly reduced when the wheat was subjected to heat stress, while malonaldehyde content increased, and the enzyme activities were significantly enhanced when KH_2_PO_4_ was sprayed. Heat stress significantly decreased the contribution rate of dry matter accumulation (DM) after anthesis of wheat to grain (CRAA), whereas spraying KH_2_PO_4_ significantly increased the CRAA and harvest index. At maturity, the DM in CK was significantly higher than that in HT, KDP was significantly higher than PHT, and KDP had the highest DM. Compared with CK, the GY in KDP significantly increased by 9.85% over the two years, while the GY in HT decreased by 11.44% compared with that of CK, and the GY in PHT decreased by 6.31% compared to that of KDP. Spraying KH_2_PO_4_ after anthesis primarily helped GY by maintaining a high thousand grain weight to lessen the negative effects of heat stress on wheat. Moreover, heat stress significantly reduced protein concentration, wet gluten content, dough development time, and hardness index in grains of mature, while spraying KH_2_PO_4_ maintained a sufficient grain quality under the conditions of achieving higher yields. Overall, spraying KH_2_PO_4_ after anthesis could enhance the heat stress resistance of wheat and maintain the photosynthetic capacity of flag leaves, ensuring the dry matter production and reducing the negative effects on grain yield and quality in the 3HP.

## 1. Introduction

As one of the most extensively grown cereal crops in the world, wheat provides vital nutrients and calories [1]. It is also one of the main food crops in China, and its stable and high production is directly tied to the safety of the country’s food supply. To attain a high yield, wheat must meet strict standards for its growth environment in addition to genetic elements. Temperature, as an important climate factor, has an important impact on the growth and development of wheat [2]. Although wheat grain yield (GY) is significantly impacted by high temperature stress at several growth phases, the largest reduction in GY occurs when there is heat stress during the grain filling period [3,4]. Unfortunately, a previous study predicted that the global temperature will rise by 1.8–4.0 °C by the end of the twenty-first century [5]. Additionally, the frequency and severity of heat stress on wheat’s growth and development are rising as a result of the increase in extreme weather events [6,7]. Niwas and Khichar (2016) reported that the damage temperature threshold for wheat during the grain filling period is 35 °C [8]. Additionally, the heat stress during the filling stage has a negative impact on the grain production and quality of wheat [9,10]. However, the Huang-Huai-Hai Plain (3HP), as the largest production base of wheat in China, has recently experienced a lot of natural high temperature weather (>35 °C) during the grain filling period in recent years, which poses a serious threat to the region’s cereal production [11]. It has been demonstrated that heat stress during the filling stage of wheat drastically reduced the leaf chlorophyll content and limited the plant’s ability to assimilate carbon, causing premature senescence and a drop in production [12,13]. Additionally, heat stress enhanced the transpiration of plants, leading to a water deficit in leaves, which affected the normal physiological activities of the leaves and reduced the duration of grain filling [14,15]. Thus, it is crucial to use some practical agricultural management strategies to address the negative impacts of heat stress on wheat production that occurred during the filling period in the 3HP.

Grain weight is determined by the remobilization of stored carbohydrates in vegetative organs before anthesis of wheat and the photoassimilate in these organs during the grain filling period. One important strategy for increasing wheat yields is to increase photosynthesis in leaves during the filling period and improve the contribution rate of dry matter accumulation after anthesis to grain yield [16]. To sustain photosynthesis, a high chlorophyll concentration in the leaves must be maintained during the growing season, and the photosynthesis during the crop growing season is a major determinant for wheat biomass [17]. Earlier research had suggested that the wheat flag leaf is essential for the dry matter accumulation after anthesis and grain filling [18]. Some studies have shown that the contribution rate of dry matter accumulation after anthesis to grain yield reached more than 80% in high yielding wheat, and increasing dry matter accumulation after anthesis is an important means to improve grain weight and production [19,20]. However, in addition to the chlorophyll content of flag leaves during the period when wheat grains are filled, other environmental parameters, such as soil moisture and canopy temperature, had a significant impact on leaf photosynthesis [16,21,22]. Indeed, grain filling is extremely sensitive to heat stress, and this process would be sped up and last for less time when wheat suffered from high temperatures during the filling period [3]. Production practice has shown that after wheat anthesis in the 3HP region, the air temperature rises gradually, making it easy to experience heat stress throughout the grain filling stage. Additionally, research revealed that when wheat was exposed to heat stress during the rapid grain filling stage, the average filling rate dramatically decreased, which resulted in the early termination of grain filling and significantly lower grain weights and yields [23]. Additionally, carbon and nitrogen are the key constituents of wheat grain, and sufficient non-structural carbohydrates (NSCs) in the vegetative organs during the filling period is helpful to increase grain weight [24]. According to Yang et al. (2004), sucrose and soluble sugars make up the majority of the NSCs stored in the vegetative organs of wheat [25].

The antioxidant system in plants can keep the reactive oxygen species (ROS) in a dynamic balance throughout the plant’s normal growth and development period through a series of physiological responses for stress resistance. In plant vacuoles, ROS in the forms of hydrogen peroxide, superoxide anions, hydroxyl radicals, and singlet oxygen are present at lower levels when the temperature is suitable for the plant. However, when the plants are exposed to an aberrant external environment, they interfere with the metabolism of ROS and cause crops to produce excessive amounts of harmful substances [26]. The wheat flag leaf is crucial for the assimilation of carbohydrates and the antioxidant defense system against heat stress [27]. Previous research results have shown that the activities of superoxide dismutase (SOD), peroxidase (POD), and catalase (CAT) in wheat plant were increased with exposure to heat stress [28]. The extreme heat stress will increase ROS levels and have a negative impact on cell metabolism [29], alter cellular components, and increase cell membrane permeability, allowing hazardous substances to enter cells while decreasing the activity of plant-related antioxidant enzymes including SOD, POD, and CAT [3,30,31]. According to studies, heat stress significantly decreased the antioxidant enzyme activities in flag leaves, decreased the cellular antioxidant capacity, led to a significant accumulation of malondialdehyde (MDA), stimulated leaf aging, and resulted in lower grain weights [32]. Thus, it is crucial to take action to lessen the effects of heat stress during the filling stage on grain yield by raising the activity of antioxidant enzymes in wheat leaves and lowering the MDA levels.

Reasonable cultivation measures are important to improve the stress resistance of wheat and achieve a high yield. To avoid a meteorological calamity like heat stress during the critical period of wheat development, the date of wheat sowing, for instance, might be rationally determined based on the climate, season, and other ecological parameters [33]. Optimizing water and fertilizer management is another critical aspect in controlling wheat growth, enhancing its resilience to adversity and reducing the negative impacts of heat stress at harvest on the yield [34,35]. Exogenous mineral fertilizer administration can also greatly increased crop resistance to abiotic stress, which is a financially viable strategy to reduce the impacts of terminal heat stress on wheat production [36,37]. Potassium dihydrogen phosphate (KH_2_PO_4_), as a high-quality and effective common foliar fertilizer, plays an important role in crop stress resistance and yield improvement [38,39]. In summary, the negative impacts of heat stress on wheat grain yield and quality can be lessened by monitoring for high temperature meteorological disasters and implementing specific remedies in the later stages of wheat growth.

Based on the previous findings, spraying KH_2_PO_4_ can be used as an effective means to deal with heat stress during the wheat grain filling period. However, it is unclear how spraying KH_2_PO_4_ will affect the physiological traits of flag leaves, the accumulation and remobilization of dry matter, and grain yield and quality when exposed to heat stress during the filling period. In this study, we hypothesized that spraying KH_2_PO_4_ solution could enhance the resistance of wheat to heat stress during the grain filling period of wheat, delay leaf senescence, increase the yield, and improve the grain quality. Therefore, the primary objectives of our study were to (1) explore the effects of spraying KH_2_PO_4_ on flag leaf chlorophyll content, Pn, activities of antioxidant enzymes, and MDA content under heat stress during the filling period; (2) clarify the effects of different treatments on wheat dry matter accumulation, and grain yield and quality; (3) analyze the mechanism underlying the effects of KH_2_PO_4_ on flag leaf photosynthetic physiology, material production, and yield and grain quality. The purpose of this study is to provide a theoretical basis and technical support for the extension and application of KH_2_PO_4_ to cope with heat stress in field wheat production in the 3HP.

## 2. Results

### 2.1. Chlorophyll Content in Flag Leaf

In both years, KH_2_PO_4_ sprayed on the third and eleventh days after wheat anthesis significantly increased the chlorophyll content in the flag leaf (Figure 1). Before heat stress, compared with spraying the water treatment (CK), the spraying KH_2_PO_4_ treatment after anthesis (KDP) increased the chlorophyll content by 3.33% and 3.37% in the 2020–2021 and 2021–2022 growing seasons of wheat, respectively. After the wheat plants were subjected to heat stress, compared to CK, the chlorophyll content was significantly reduced in HT, and the chlorophyll content of KDP was significantly higher than that of PTH. However, the highest chlorophyll content was maintained in KDP after heat stress in both years. After heat stress, the decrease in percentage of chlorophyll content after spraying KH_2_PO_4_ was less than that in the non-sprayed treatment. Compared with CK, the chlorophyll content decreased by 2.37% and 4.48% in HT in the two growing seasons, respectively, and by 2.07% and 4.40% in PHT compared to the KDP treatment in both years, respectively. There was no significant difference in the chlorophyll content between PHT and CK in the 2020–2021 growing season of wheat, while it was significantly increased in PHT compared to CK in the second growing season. The aforementioned findings show that spraying KH_2_PO_4_ could increase the chlorophyll content of wheat flag leaves throughout the grain filling period and sustain the greater chlorophyll content when heat stress occurs.

### 2.2. Net Photosynthetic Rate in Flag Leaf

Figure 2 illustrates that spraying KH_2_PO_4_ after anthesis significantly increased the net photosynthetic rate (Pn) in flag leaves, the heat stress significantly decreased the Pn compared to CK, and the Pn of KDP was significantly higher than that of PHT. According to these findings, heat stress reduced photosynthetic ability during the grain filling period, which may have contributed to the considerable drop in chlorophyll concentration observed in wheat after exposure to heat stress (Figure 1). However, after heat stress, the Pn of HT decreased by 11.78% and 1.16% in 2020–2021 and 2021–2022, respectively, compared with that of CK, while compared with KDP, the Pn in PHT decreased by 8.93% and 5.94%, respectively. In summary, spraying KH_2_PO_4_ after anthesis could sustain the greater photosynthetic capacity of flag leaves when heat stress occurred and considerably increased the net photosynthetic rate of flag leaves during the grain filling period.

### 2.3. Antioxidant Enzyme Activities and MDA in Flag Leaves

The antioxidant enzyme activities in leaves are closely related to the crop’s ability to withstand environmental stress, and the concentration of MDA reflects the extent of cell membrane damage. The findings demonstrated that spraying KH_2_PO_4_ significantly boosted the enzyme activities of superoxide dismutase (SOD), peroxidase (POD), and catalase (CAT) in flag leaves and lowered the malonaldehyde (MDA) content (Figure 3). The enzyme activities of SOD, CAT, and POD were significantly lower in HT than CK, and they were significantly lower in PHT than PT after the wheat were subjected to heat stress during the grain filling period. However, HT had a significantly higher MDA content than CK, and PHT had a significantly higher MDA content than KDP. However, there was no significant difference in the enzyme activities of SOD and POD in flag leaves after heat stress between PHT and CK, and PHT had a significantly higher CAT activity than CK. PHT had a significantly higher MDA content, indicating that spraying KH_2_PO_4_ could improve wheat’s resistance to heat stress during the filling period.

### 2.4. Dry Matter Production of Wheat

#### 2.4.1. Dry Matter Accumulation and Remobilization Pre and Post Anthesis

Across the two years, the dry matter accumulation of the wheat population at maturity (DMM) in CK was significantly higher than in HT, and it was significantly higher in KDP than PHT, with KDP achieving the highest DMM (Table 1). Compared with HT, DMM in CK, PHT, and KDP increased by 7.74%, 9.50%, and 14.50%, respectively. In the 2020–2021 growing season, there was no significant difference in DMM between PHT and CK, but in the 2021–2022 growing season, PHT had a much greater DMM than CK. In the first growing season, the dry matter remobilization before anthesis to grain (DMR) in HT was the highest among all treatments, and there was no significant difference between PHT and CK; KDP had the lowest DMR. In the second growing season, there was no significant difference in DMT between KDP and PHT, which were significantly lower than those of CK and HT, with HT having the highest DMT. The variation in contribution rate of the dry matter remobilization to grain yield (CRBA) were similar to those for DMR among the different treatments over the two years. However, the changes in contribution rate of post-anthesis DM among the different treatments was opposite to those of CRBA. As for the harvest index (HI), in the 2020–2021 growing season KDP had a significantly higher HI than the other treatments, and there was no significant difference between PHT and CK, and they were significantly higher than HT. The HI of HT was the lowest during the growing season of 2021–2022, even though there was no significant difference in HI between KDP and PHT, and both HI values were significantly higher than that of CK. Overall, the studies above showed that spraying KH_2_PO_4_ after anthesis may significantly increase the CRAA and HI, and reduce the negative impact of heat stress on dry matter production.

#### 2.4.2. Dry Matter Accumulation and Partitioning of Wheat Plants at Maturity

Table 2 shows that the dry matter accumulation (DM) of the stem, sheath, leaf, and spike in the growing season of 2020–2021 was significantly decreased in HT compared to CK, and there was no significant difference between CK and PHT; KDP had the highest DM values. Additionally, the DM in CK, PHT, and KDP increased by 14.24%, 14.67%, and 24.80%, respectively, compared with that of HT. The ratio of stem and sheath to total DM in HT was significantly increased compared with those of CK and PHT, and there was no significant difference between the latter two; KDP had the lowest ratio of stem and sheath to total DM. There was no significant difference in the ratio of leaf to total DM among the different treatments in this year. The ratio of spike to total DM in CK was significantly larger than that of HT, while KDP had a higher ratio than PHT, and there was no significant difference between CK and PHT. In the 2021–2022 growing season, the DM of the stem and sheath in HT was the lowest among all the treatments, and there was no significant difference between KDP and PHT, which were significantly higher than that of CK. However, there was no significant effect of heat stress or spraying KH_2_PO_4_ on the ratio of the stem and sheath to total DM. HT had the highest stem and sheath to total DM ratio. The highest DM of leaves was in the order of KDP > PHT > CK > HT. Additionally, the ratios of leaf to total DM in KDP and PHT were significantly higher than CK; HT had the lowest ratio. There was little difference in the DM of the spike between CK, KDP, and PHT, and, they were significantly higher than that of HT. Compared with HT, the DM in CK, PHT and KDP increased by 7.18%, 9.45%, and 11.36%, respectively. Additionally, there was no significant difference in the ratio of spike to total DM among all the treatments in this year. In conclusion, the above results indicated that heat stress during the filling period significantly reduced the DM of the different organs in wheat, while spraying KH_2_PO_4_ after anthesis could reduce the adverse effects of heat stress on the dry matter production of wheat.

### 2.5. Grain Yield and Yield Components at Harvest

The ANOVA results in Table 3 show that year type (Y), spraying KH_2_PO_4_ (P), and heat stress (H) had significant effects on grain yield (GY) and thousand grain weight (TGW) of wheat, and there was no significant interaction between the factors on the GY; however, the TGW was significantly affected by Y × P, P × T, and Y × P × H. The spike number (SN) was only significantly affected by Y. Across the two years, the GY in KDP was significantly higher than that of PHT and CK, and there was no obvious difference between the latter two in GY; HT had the lowest GY (Table 4). Compared with CK, in the 2020–2021 and 2021–2022 growing seasons, the GY of KDP was increased by 10.89% and 8.81%, respectively; the PHT treatment decreased the GY by 6.18% and 6.44% compared to that of KDP, respectively. No significant difference was observed in SN and grain number (GN) among the different treatment. The TGW of KDP was significantly higher than that of PHT, which was higher than that of CK; HT had the lowest SN and GN values. Overall, spraying KH_2_PO_4_ after anthesis could significantly improve wheat yield, and reduce the adverse effects of heat stress on yield by maintaining the grain weight.

### 2.6. Soluble Sugar and Sucrose Content in Stems and Sheaths and Flag Leaves

As showed in Figure 4, compared with the spraying water treatment, the soluble sugar (SS) and sucrose (SU) contents in flag leaves, stems and sheaths were significantly improved with the spraying KH_2_PO_4_ treatment. After heat stress, the contents of SS and SU in the two organs were significantly higher in KDP than PHT. The SS and SU contents of PHT were significantly higher than CK, and HT had the lowest contents. Compared with HT, the content of SS and SU in flag leaves of CK, KDP, and PHT increased by 8.00%, 24.08%, and 15.49% and 9.17%, 26.60, and 17.76%, respectively. Additionally, in the stem and sheath, the contents were increased by 7.72%, 23.19%, and 15.69% and 9.69%, 19.30%, and 13.44%, respectively. Overall, spraying KH_2_PO_4_ prevent the vegetative organs’ sugar levels from dropping during the grain filling period.

### 2.7. Grain Quality Parameters after Harvest

The ANOVA results in Table 3 show that the grain protein content (PC), test weight (TW), wet gluten content (WGC), development time (DT), and hardness index (HAI) were significantly affected by Y, while DT and HAI only were significantly affected by P. Except for DT, all the grain quality parameters were significantly affected by T. HAI was affected by Y × P and Y × T. Across the two years, there was little difference in PC and WGC between the spraying water and KH_2_PO_4_ treatment (Table 5). Compared with CK, the PC and WGC in HT decreased by 5.33%, 2.77% and 4.714%, 2.89%, respectively. Additionally, the PC and WGC decreased by 4.58%, 3.38% and 3.99%, 3.54% in PHT compared to KDP, respectively. No significant difference was observed in the TW among the different treatments. Heat stress significantly decreased the DT and HAI, and there was no significant difference between spraying water and KH_2_PO_4_ in the 2020–2021 growing season. However, the DT in HT was significantly lower than that of PHT in the 2021–2022 growing season, and the HAI in CK was significantly lower than that of KDP in this year. Overall, spraying KH_2_PO_4_ after anthesis could significantly improve the grain quality of wheat, and spraying KH_2_PO_4_ could reduce the adverse effects of heat stress on grain quality.

## 3. Discussion

### 3.1. Effects of Spraying KH_2_PO_4_ on Photosynthesis in Flag Leaves of Wheat under Heat Stress during the Filling Period

One of the most crucial metabolic processes in crops is photosynthesis. The photosynthetic organs serve as the primary source of dry matter production for crop growth and development, and the chlorophyll content in the leaves of crops directly correlates with its photosynthetic capability. However, the chlorophyll content decreases dramatically when the plant experiences heat stress, affecting plant growth, photosynthesis, metabolism, and productivity [40,41]. Previous studies have shown that heat stress significantly reduces photosynthesis because it reduces the chlorophyll levels in the leaves and decreases the efficiency of PSII, which disrupts electron transport [42]. Additionally, after heat stress during the grain filling period, the photosynthetic parameters in wheat leaves, including net photosynthetic rate (Pn), stomatal conductance, transpiration rate, photochemical efficiency, and actual photosynthetic yield, drastically decreased [43,44]. According to research by Nasehzadeh et al. (2017), heat stress that occurs during the filling period has the largest negative effects on wheat yield compared to other times, causing a considerable decrease in wheat yield [45]. Spraying chemical regulators on wheat, however, could reduce the early senescence of flag leaves brought on by heat stress, lengthen its green retention, and boost the accumulation of dry matter in grain [46]. In this present study, wheat plants sprayed with a 0.3% KH_2_PO_4_ solution after anthesis significantly improved the chlorophyll content (Figure 1) and enhanced the Pn in flag leaves before heat stress treatment, and maintained a higher Pn when subjected to high temperatures (Figure 2). The foliage spraying of KH_2_PO_4_ may supplement the necessary phosphorus (P) and potassium (K) elements for plants, which play an important role in various metabolic activities and biological pathways, such as in photosynthesis, cell expansion, osmoregulation, and stomatal movement regulation, and they also work as activation factors for several enzymes [47,48]. However, the Pn in flag leaves of the non-sprayed treatment was significantly decreased due to its significantly decreased chlorophyll contents. This means that spraying of 0.3% KH_2_PO_4_ after anthesis can effectively reduce the damaging effects of heat stress on the photosynthetic capacity of wheat leaves during the filling period. This is consistent with the previous findings that spraying KH_2_PO_4_ could increase the chlorophyll content in flag leaves, increase grain weight, and reduce the adverse effects on wheat yields [46]. Spraying KH_2_PO_4_ on the leaves could alleviate the heat damage caused by high temperatures during the filling period. This effect may be related to the potential role of potassium in the synthesis of photosynthesis-related enzymes in the leaves, which would enhance photosynthesis [49], but the further research into the physiological mechanism is required.

### 3.2. Effects of Spraying KH_2_PO_4_ on the Antioxidant Enzyme Activity of Wheat Flag Leaves under Heat Stress during the Filling Period

By activating a large number of stress-responsive genes and synthesizing a variety of functional proteins through a complex signal transduction network, plants have evolved a range of physiological and metabolic responses to cope with unfavorable conditions in order to confer tolerance to environmental stresses [41,50]. However, when the crop is exposed to heat stress, the stress causes physiologic changes in plant cells and disordered in the antioxidant system [30]. This results in a large buildup of MDA, which damages the cell stability and structure, significantly lowers the photosynthetic ability of flag leaves [51,52,53]. Nevertheless, the crops will immediately begin the necessary defense mechanisms to withstand that adversity when it occurs, minimizing any potential negative effects of adversity [54,55]. However, if the stress is too extreme or is prolonged, the related enzyme activities may be adversely affected, lowering the effectiveness of its defense mechanisms and leading to oxidative stress reactions in the crops [56]. Our study results found that wheat subjected to heat stress during the grain filling period significantly increased the MDA content in flag leaves (Figure 3). This may due to the excessive accumulation of ROS in the plants, causing lipid peroxidation, damaging nucleic acids, and oxidizing proteins [57]. Although spraying a 0.3% KH_2_PO_4_ solution after anthesis also improved the MDA content in flag leaves, it was significantly lower than the non-sprayed treatment. In this study, the enzyme activities of SOD, POD, and CAT in plants sprayed with a 0.3% KH_2_PO_4_ solution after anthesis were significantly higher than those of the non-sprayed treatment, which explained why the MDA content was lower with the 0.3% KH_2_PO_4_ solution treatment. Spraying 0.3% KH_2_PO_4_ may prevent ROS accumulation, and the negative effect of ROS can be eliminated by enhancing antioxidant enzyme activities in flag leaves, preventing oxidative damage under heat stress [58]. Additionally, this is consistent with a previous study [46]. The spraying of KH_2_PO_4_ on the leaves may supplement them with the phosphorus and potassium needed by the plant, enhancing the stability of the leaf structure, and allowing them to respond more quickly to the heat stress. This would improve the plant’s ability to reduce the harm from the heat stress. As reported by Lv et al. (2017), phosphorus can increase the activities of antioxidant enzymes and decrease the MDA content in flag leaves in the filling period, thus delaying the senescence of wheat [38].

### 3.3. Effects of Spraying KH_2_PO_4_ on the Dry Matter Production of Wheat under Heat Stress during the Filling Period

The grain yield mainly derives from the dry matter remobilization accumulated in wheat vegetative organs before anthesis and the dry matter accumulation (DM) produced by the photosynthetic organs after anthesis; thus, increasing DM post-anthesis is an important means to increase crop grain yield [19]. Research has shown that the DM in the stem and sheath was significantly decreased under heat stress after anthesis, reducing the translocation amount from the vegetative organs to the spike, thus reducing the grain yield (GY) [59]. However, enhancing the photosynthetic performance of wheat leaves during the grain filling period is the basis for improving dry matter production and achieving a high GY [60]. In this study, heat stress significantly decreased the total dry matter accumulation at maturity (DMM) of wheat, and reduced the contribution ratio of dry matter accumulation post-anthesis to grain yield (CRAA) and harvest index (HI). However, spraying a 0.3% KH_2_PO_4_ solution significantly increased the DMM compared with the water treatment under heat stress, and improved the CRAA and HI. This means that spraying the KH_2_PO_4_ after anthesis of wheat can delay plant senescence and alleviate the harm of heat stress on the photosynthetic capacity of the leaves and promote dry matter production. The reason for the delayed plant senescence is that spraying KH_2_PO_4_ significantly enhanced the enzyme activities of SOD, POD, and CAT in flag leaves under heat stress (Figure 3), and maintained a higher Pn (Figure 2). Moreover, spraying KH_2_PO_4_ reduced the negative effects of heat stress on grain weight and increased GY, which is consistent with the results of a previous study [46]. Prior research indicated that P has a structural and functional role in the photosynthetic carbon cycle’s activated intermediates, energy metabolites, membrane lipids, and nucleic acids; in addition, inorganic phosphate is essential for signal transduction cascades [61]. Additionally, the foliage application of K intensifies the carbohydrate transport from the source to the sink [38]. In this present study, the KH_2_PO_4_, as a phosphate and potassium compound fertilizer, had a positive role in enhancing the photosynthetic capacity and stress resistance of wheat plants [48]. It was also conducive to promoting the transport of photosynthetic substances to the grain, which may provide more favorable conditions for the transport of substances when the crop was subjected to heat stress in this study, thereby reducing the detrimental impact on GY.

### 3.4. Effects of Spraying KH_2_PO_4_ on the Grain Weight and Quality of Wheat under Heat Stress during the Filling Period

The critical growth period for the production of grain weight in cereal crops is known as grain filling, and the main parameters impacting grain weight are the filling rate and duration [62]. Many studies have demonstrated that stresses such extreme heat or drought will reduce the average rate of grain filling and cause a decline in GY [63,64,65]. However, the grain filling rate could be prolonged by spraying mineral nutrition in the late growth stage of wheat to improve crop yield [36,38]. In our investigation, heat stress during the grain filling stage greatly lowered the thousand grain weight (TGW) and GY; however, spraying KH_2_PO_4_ after anthesis counteracted this impact (Table 4), which may be related to potassium’s ability to improve the activity of starch synthase and promote sugar metabolism in wheat [66]. According to previous studies, P and K are significant dietary components that have a considerable impact on wheat grain weight [67], and spraying KH_2_PO_4_ after anthesis has been shown to greatly accelerate grain filling and increase grain weight [68]. Moreover, maintaining high NSC levels in vegetative organs during wheat filling is advantageous for promoting grain filling and raising grain weight [69,70]. In this study, we discovered that spraying KH_2_PO_4_ after anthesis preserved a high level of NSC content in the vegetative organs after heat stress, while it was considerably reduced in the non-sprayed plants (Figure 4). Additionally, it provided the material and energy basis for grain filling in PHT, which could explain why spraying KH_2_PO_4_ reduced the negative effects on TGW and GY due to the heat stress during the grain filling period in this present study.

Heat stress affects the grain quality of many cereal crops mainly though limiting the assimilation and transport of nutrients to the grain in vegetative organs [23]. Additionally, the grain quality of wheat is significantly affected by heat stress during the grain filling period [71,72]. Dias et al. (2008) reported that the increase in grain protein content was related to the sedimentation index and intensity of essential amino acids in grain, and they were significantly decreased when subjected to heat stress [73]. In this study, the heat stress significantly reduced the PC, WGC, and FT of grain after heat stress during the filling period (Table 5). However, there was little difference in the grain quality between KDP and CK, and between HT and PHT, which may be due to the high grain yield obtained by KDP and PHT compared with that of CK and HT. For wheat production, it is always challenging to coordinate high yields and good quality, and high yield always results in a deterioration in grain quality [74]. According to this study, spraying KH_2_PO_4_ could maintain grain quality under high GYs, which may be related to the fact it may promote the transport of amino acids from vegetative organs to wheat grain as well as the accumulation of nitrogen and protein content in the grain. However, the mechanism through which spraying KH_2_PO_4_ achieved a high GY and maintained the grain quality under heat stress needs to be studied further.

## 4. Materials and Methods

### 4.1. Experimental Site

From 2020 to 2022, during the winter wheat growing seasons, an in situ field experiment was conducted in the Agricultural Extraction Garden of Anhui Agricultural University (31°86′ N, 117°26′ E) in Hefei City, Anhui Province, China. This experimental site has an altitude of 30 m. The average annual temperature in this region, which has a subtropical monsoon climate, is approximately 15 °C. The total rainfall was 391.5 mm and 387.1 mm during the 2020–2021 and 2021–2022 growing seasons of wheat, respectively. Before planting wheat, the experimental field fertility in the top 20 cm of the soil was analyzed. The results showed that there was 14.3 g·kg^−1^ of organic matter content, 101.5 mg·kg^−1^ of total nitrogen, 43.9 mg·kg^−1^ of available phosphorus, and 314.0 mg·kg^−1^ of potassium.

### 4.2. Experimental Design

In this field experiment, foliar spraying water combined with heat stress (HT), 0.3% KH_2_PO_4_ (KDP), and 0.3% KH_2_PO_4_ combined with heat stress (PHT) were designed, and spraying water alone was used as a control (CK). KH_2_PO_4_ was purchased from Beijing Solarbio Science & Technology Co., Ltd. (China, Beijing). Three replicates were used in the randomized complete block experimental design. The area of each experimental plot was 9 m^2^ (3 m × 3 m). A high-yield winter wheat cultivar named ‘Annong 0711’, one of the most widely cultivated cultivars in the south of the 3HP, was chosen for this experiment. Wheat plants were sprayed on the third and eleventh days after anthesis, and the spraying time was chosen to be after 16:00 p.m. on a day with no wind for better absorption and uniform application of the KH_2_PO_4_ solution to the crop. A vapor pressure-type sprayer was used to apply 45 g·m^−2^ of a 0.3% KH_2_PO_4_ solution to KDP and PHT each time according to Cao et al. [39], and the same quantity of water was applied to CK and HT. A plastic shed was used to simulate the effects of heat stress in the field. The first year’s heat stress time was from the 18th to 22nd after wheat anthesis, while the second year’s heat stress time was delayed by two days, from the 20th to 24th, due to the occurrence of overcast weather. The daily heat stress time was from 11: 00 a.m. to 16: 00 p.m. Inside and outside the shed, thermohygrometers (RC-4, Jing Chuang, China) were mounted, and the air temperature was automatically recorded every hour. The temperatures inside and outside of the shed are displayed in Figure 5. No rainfall occurred throughout the duration of the simulated heat stress during the two years, and the wheat did not experience any drought stress from anthesis to maturity. According to the local fertilizer application schedule for wheat production, basal fertilizers consisting of 105 kg·ha^−1^ nitrogen (N), 112.5 kg·ha^−1^ P_2_O_5_, and 112.5 kg·ha^−1^ K_2_O were applied before sowing seeds. All experimental plots were artificially applied with 90 kg·ha^−1^ N at the jointing stage of wheat. At a planting density of 180 plants per square meter, the wheat seedlings were sown on 30 October 2020 and 30 October 2021, and were harvested on 25 May 2021 and 24 May 2022, respectively.

### 4.3. Sampling and Measurements

#### 4.3.1. Net Photosynthetic Rate (Pn)

A portable photosynthetic measurement instrument, the Li-6400 (LI-COR, Lincoln, NE, USA), was used to determine the Pn of wheat flag leaves in the main stem before and after heat stress treatment. In each experimental plot, three leaves were randomly chosen and measured (9:00–11:00 a.m.) for each treatment.

#### 4.3.2. Chlorophyll Content, Antioxidant Enzyme Activities, and MDA Content

Before and after the heat stress treatment, twenty flag leaves were randomly sampled from each experimental plot and divided into four equal parts to analyze the chlorophyll content (a + b), antioxidant enzyme activities, and MDA content. The samples were immediately sealed in tinfoil, frozen using liquid nitrogen, and stored in an ultra-low temperature box at −80 °C until the measurement. For chlorophyll content, the leaf samples were cut into small pieces on ice, promptly weighed, and then extracted for 48 h in the dark using 95% ethanol. After extraction, optical density (OD) values at 649 and 665 nm were measured using a spectrophotometer [20]. Following the procedures described by Yang et al. (2007) [75] and Zheng et al. (2009) [76], the enzyme activities of SOD, CAT, and POD, and the MDA content in flag leaves were measured.

#### 4.3.3. Soluble Sugar and Sucrose Content

The flag leaf, stem, and sheath of ten plants of wheat were removed from each experimental plot before and after the heat stress, and all samples were dried in an oven at 75 °C to a constant weight. The soluble sugar and sucrose content in these samples were measured according to the methods described by Winter and Huber (2000) [77].

#### 4.3.4. Dry Matter Accumulation and Grain Yield (GY)

At the anthesis and maturity stages of wheat, samples of two 50 cm wheat plants from the inner rows of each experimental plot were removed at ground level and divided into stem and sheath, leaf, spike, and grain. All samples were dried to a consistent weight in an oven at 75 °C. According to Shi et al. (2016) [16,19], the difference between the dry matter accumulation (DM) at anthesis and the DM at maturity without grain weight was used to calculate the dry matter remobilization (DMR) from the vegetative organs to the grain between anthesis and maturity. Post-anthesis DM was calculated as the difference between grain weight and DMR. The contribution rate of pre-anthesis DM to grain was calculated from the ratio of DMR to grain yield at maturity (CRBA), and the contribution rate of post-anthesis DM to grain yield was calculated using 100-CRBA (CRAA). The harvest index (HI) was calculated as the ratio of grain to all accumulated above-ground dry matter at harvest. Spikes were counted in each plot prior to harvest to determine the spike number. Fifty randomly selected spikes from each experimental plot were counted to calculate the grain number. One square meter of wheat plants from each experimental plot were harvested at maturity, and the grain yield was calculated after artificial threshing. Actual grain yield was reported on a 13% moisture basis. The thousand grain weight was calculated by weighing 1000 seeds from each sample and averaging the values of three replicates.

#### 4.3.5. Grain Quality Parameters

At maturity, some wheat spikes were randomly selected from each experimental plot and manually threshed, and were then dried naturally to determine grain quality parameters. Three replicates were analyzed for each experimental treatment. When measuring the samples, approximately 100 g of wheat grains were taken and placed in a sample cup, and the grain protein content, test weight, wet gluten content, hardness index, and development time was measured using a near infrared analyzer (Perten DA7200, Sweden), which purchased from Perten Instruments Co., Ltd. (Beijing, China).

### 4.4. Data Analysis

SPSS Statistics 22.0 software (IBM, Armonk, NY, USA) was used to analyze the data, and the least significant difference test (*p* = 0.05) was used to compare the difference between different treatments in this study. All figures in this paper were generated using OriginPro 2019 (OriginLab Corp., Northampton, MA, USA).

## 5. Conclusions

Spraying KH_2_PO_4_ could increase the chlorophyll content and net photosynthetic rate in flag leaves under heat stress during the grain filling period by enhancing the activities of SOD, CAT, and POD and decreasing the MDA content in the flag leaf, resulting in promoting dry matter accumulation after the anthesis of wheat. It also increased the NSC content in the flag leaf and stem and sheath by spraying KH_2_PO_4_, causing an improvement of grain yield and maintenance of the high grain quality.

## Figures and Tables

**Figure 1 plants-12-01801-f001:**
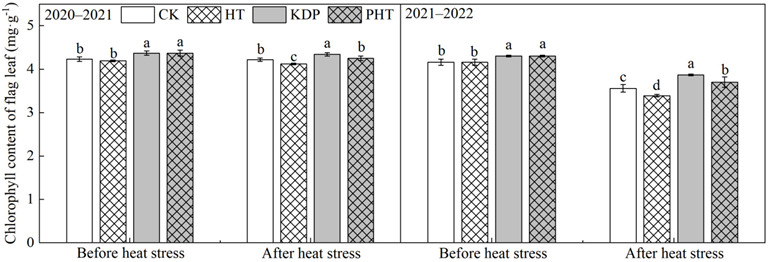
Effects of spraying KH_2_PO_4_ on chlorophyll content in flag leaf of wheat under heat stress during the grain filling period in the 2020–2021 and 2021–2022 growing seasons. CK, HT, KDP, and PHT represent spraying water, water combined with heat stress, 0.3% KH_2_PO_4_, and 0.3% KH_2_PO_4_ combined with heat stress, respectively. Different lowercase letters indicate significant differences between treatments (*p* < 0.05). Vertical bars represent the standard error of the mean.

**Figure 2 plants-12-01801-f002:**
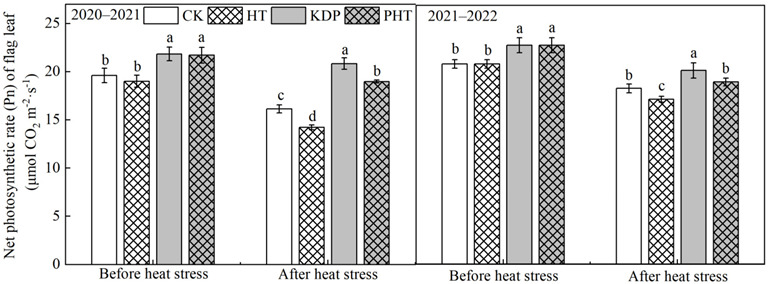
Effects of spraying KH_2_PO_4_ on net photosynthetic rate (Pn) in flag leaf of wheat under heat stress during the grain filling period in the 2020–2021 and 2021–2022 growing seasons. CK, HT, KDP, and PHT represent spraying water, water combined with heat stress, 0.3% KH_2_PO_4_, and 0.3% KH_2_PO_4_ combined with heat stress, respectively. Different lowercase letters indicate significant differences between treatments (*p* < 0.05). Vertical bars represent the standard error of the mean.

**Figure 3 plants-12-01801-f003:**
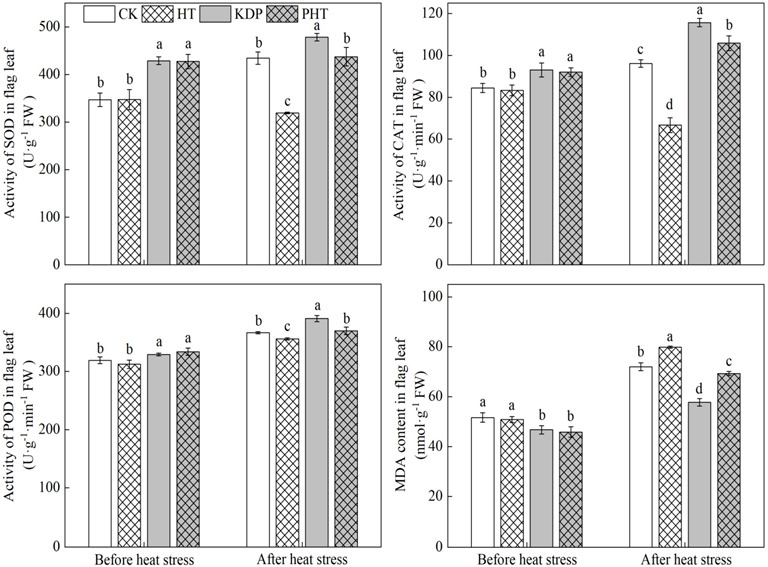
Effects of spraying KH_2_PO_4_ on protective enzyme activity and MDA content in flag leaf of wheat under heat stress during the grain filling period in the 2021–2022 growing season. CK, HT, KDP, and PHT represent spraying water, water combined with heat stress, 0.3% KH_2_PO_4_, and 0.3% KH_2_PO_4_ combined with heat stress, respectively. Different lowercase letters indicate significant differences between treatments (*p* < 0.05). Vertical bars represent the standard error of the mean.

**Figure 4 plants-12-01801-f004:**
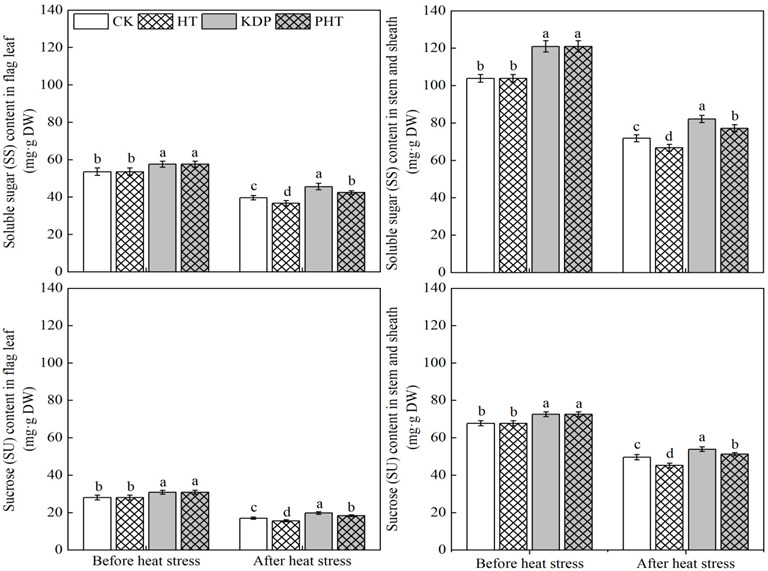
Effects of spraying KH_2_PO_4_ on soluble sugar and sugar content in flag leaf and stem and sheath of wheat under heat stress during the grain filling period in the 2021–2022 growing season. CK, HT, KDP, and PHT represent spraying water, water combined with heat stress, 0.3% KH_2_PO_4_, and 0.3% KH_2_PO_4_ combined with heat stress, respectively. Different lowercase letters indicate significant differences between treatments (*p* < 0.05). Vertical bars represent the standard error of the mean.

**Figure 5 plants-12-01801-f005:**
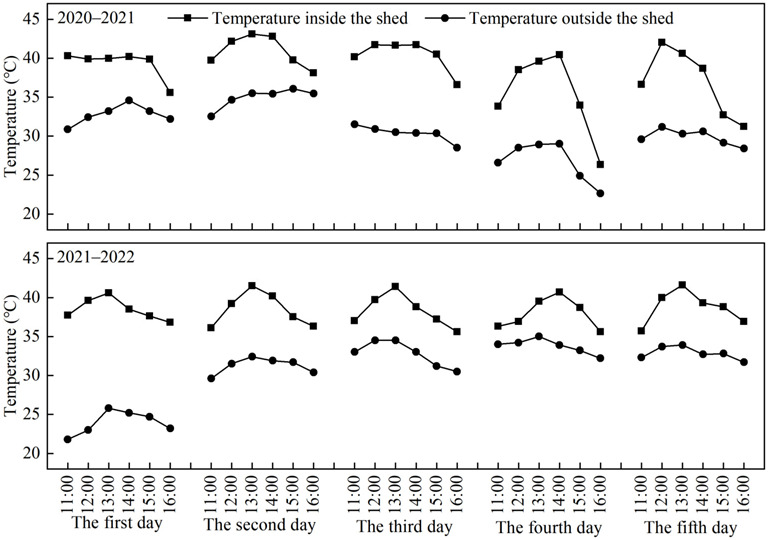
Temperature both inside and outside of the shed during heat stress period in this study.

**Table 1 plants-12-01801-t001:** Effects of spraying KH_2_PO_4_ on wheat dry matter accumulation, remobilization and harvest index under heat stress during the grain filling period.

Year	Treatment	DMM (t·ha^−1^)	DMR (t·ha^−1^)	CRBA (%)	CRAA (%)	Harvest Index (%)
2020–2021	CK	22.61 ± 0.03 b	4.62 ± 0.07 b	42.35 ± 0.55 b	57.65 ± 0.55 b	48.30 ± 0.14 b
	HT	20.23 ± 0.23 c	5.63 ± 0.15 a	58.94 ± 1.64 a	41.06 ± 1.64 c	47.23 ± 0.44 c
	KDP	24.48 ± 0.54 a	3.94 ± 0.18 c	32.61 ± 2.38 c	67.39 ± 2.38 a	49.47 ± 0.31 a
	PHT	22.74 ± 0.33 b	4.51 ± 0.21 b	41.26 ± 2.64 b	58.74 ± 2.64 b	48.12 ± 0.07 b
2021–2022	CK	20.93 ± 0.10 c	2.63 ± 0.12 b	27.71 ± 1.22 b	72.29 ± 1.22 b	45.41 ± 0.34 b
	HT	20.18 ± 0.30 d	2.92 ± 0.10 a	32.33 ± 1.82 a	67.67 ± 1.82 c	44.78 ± 0.34 c
	KDP	21.83 ± 0.08 a	2.37 ± 0.11 c	23.36 ± 0.98 c	76.64 ± 0.98 a	46.44 ± 0.36 a
	PHT	21.51 ± 0.12 b	2.47 ± 0.06 bc	24.92 ± 0.75 c	75.08 ± 0.75 a	46.13 ± 0.15 a

CK, HT, KDP, and PHT represent spraying water, water combined with heat stress, 0.3% KH_2_PO_4_, and 0.3% KH_2_PO_4_ combined with heat stress, respectively. DMM, dry matter accumulation at maturity; DMR, dry matter remobilization before anthesis of vegetative organs; CRBA, the contribution ratio of dry matter remobilization before anthesis to grain; CRAA, contribution ration of dry matter accumulation after anthesis to grain; HI, harvest index. Different lowercase letters following the data in the same column indicate significant differences (*p* < 0.05). Values are means ± standard error (*n* = 3).

**Table 2 plants-12-01801-t002:** Effects of spraying KH_2_PO_4_ on dry matter accumulation and distribution of wheat at maturity under heat stress during the grain filling period.

Year	Treatment	Stem and Sheath		Leaf		Spike	
		DM (t·ha^−1^)	Ratio (%)	DM (t·ha^−1^)	Ratio (%)	DM (t·ha^−1^)	Ratio (%)
2020–2021	CK	7.20 ± 0.02 b	31.83 ± 0.07 b	2.33 ± 0.02 b	10.32 ± 0.09 a	13.08 ± 0.04 b	57.85 ± 0.16 ab
	HT	6.72 ± 0.12 c	33.22 ± 0.23 a	2.06 ± 0.02 c	10.18 ± 0.12 a	11.45 ± 0.11 c	56.60 ± 0.19 c
	KDP	7.66 ± 0.18 a	31.29 ± 0.09 c	2.52 ± 0.08 a	10.31 ± 0.20 a	14.29 ± 0.30 a	58.39 ± 0.29 a
	PHT	7.24 ± 0.08 b	31.86 ± 0.48 b	2.37 ± 0.08 b	10.41 ± 0.24 a	13.13 ± 0.25 b	57.73 ± 0.49 b
2021–2022	CK	6.85 ± 0.11 b	32.62 ± 0.45 ab	2.36 ± 0.04 c	11.23 ± 0.09 b	11.79 ± 0.12 a	56.15 ± 0.41 a
	HT	6.71 ± 0.06 c	33.77 ± 0.96 a	2.17 ± 0.08 d	10.90 ± 0.21 c	11.00 ± 0.38 b	55.34 ± 0.17 a
	KDP	7.14 ± 0.05 a	32.47 ± 0.59 b	2.60 ± 0.03 a	11.81 ± 0.14 a	12.25 ± 0.26 a	55.72 ± 0.52 a
	PHT	7.00 ± 0.06 a	32.49 ± 0.38 b	2.51 ± 0.01 b	11.63 ± 0.03 a	12.04 ± 0.13 a	55.88 ± 0.39 a

CK, HT, KDP, and PHT represent spraying water, water combined with heat stress, 0.3% KH_2_PO_4_, and 0.3% KH_2_PO_4_ combined with heat stress, respectively. DM, dry matter accumulation. Different lowercase letters following the data in the same column indicate significant differences (*p* < 0.05). Values are means ± standard error (*n* = 3).

**Table 3 plants-12-01801-t003:** Analysis of variance (ANOVA) of grain yield, yield components, and grain quality parameters as affected by year type, spraying KH_2_PO_4_, and heat stress.

Factors	GY	SN	GN	TGW	PC	TW	WGC	DT	HAI
Y	2.69 *	0.28 ns	15.34 ***	36.07 ***	180.71 ***	261.26 ***	123.17 ***	55.74 ***	795.02 ***
P	51.03 ***	0.15 ns	1.00 ns	194.18 ***	0.05 ns	6.35 *	0.04 ns	3.28 ns	8.60 **
H	29.10 ***	0.00 ns	0.06 ns	111.50 ***	60.69 ***	3.91 ns	41.95 ***	65.36 ***	82.77 ***
Y × P	1.00 ns	0.53 ns	1.17 ns	6.51 *	0.01 ns	0.07 ns	0.07 ns	1.22 ns	5.43 *
Y × H	0.12 ns	0.38 ns	0.44 ns	3.40 ns	2.48 ns	0.25 ns	1.00 ns	1.22 ns	6.16 *
P × H	1.47 ns	1.13 ns	2.01 ns	17.00 ***	0.00 ns	0.66 ns	0.06 ns	0.16 ns	0.01 ns
Y × P × H	0.21 ns	0.53 ns	0.69 ns	5.36 *	0.46 ns	0.54 ns	0.14 ns	1.22 ns	0.46 ns

CK, HT, KDP, and PHT represent spraying water, water combined with heat stress, 0.3% KH_2_PO_4_, and 0.3% KH_2_PO_4_ combined with heat stress, respectively. GY represents grain yield; SN represents spike numbers; GN represents grain number per spike; TGW represents thousand grain weight; PC represents protein content; TW represents test weight; WGC represents wet gluten content; DT represents development time; HAI represents hardness index. * indicates significance at the 0.05 probability level; ** indicates significance at the 0.01 probability level; *** indicates significance at the 0.001 probability level. ns, not significant.

**Table 4 plants-12-01801-t004:** Effects of spraying KH_2_PO_4_ on yield and yield components of wheat under heat stress during the filling period.

Year	Treatment	GY (t·ha^−1^)	SN (×10^4^)	GN (Spike^−1^)	TGW (g)
2021–2021	CK	9.77 ± 0.19 b	633.33 ± 8.79 a	41.68 ± 0.42 a	43.09 ± 0.67 b
	HT	8.55 ± 0.36 c	636.10 ± 11.11 a	41.61 ± 0.59 a	39.09 ± 0.10 c
	KDP	10.83 ± 0.20 a	635.00 ± 7.24 a	41.78 ± 0.42 a	45.23 ± 0.07 a
	PHT	10.16 ± 0.46 b	628.87 ± 4.18 a	41.47 ± 0.23 a	43.96 ± 0.22 b
2021–2022	CK	10.07 ± 0.48 b	633.09 ± 2.55 a	41.92 ± 0.58 a	44.06 ± 0.29 b
	HT	9.11 ± 0.25 c	635.32 ± 3.34 a	42.47 ± 0.48 a	41.81 ± 0.78 c
	KDP	10.95 ± 0.46 a	634.76 ± 2.55 a	42.80 ± 0.49 a	46.07 ± 0.64 a
	PHT	10.25 ± 0.66 b	635.32 ± 1.67 a	42.43 ± 0.69 a	44.59 ± 0.75 b

CK, HT, KDP, and PHT represent spraying water, water combined with heat stress, 0.3% KH_2_PO_4_, and 0.3% KH_2_PO_4_ combined with heat stress, respectively. Different lowercase letters following the data in the same column indicate significant differences (*p* < 0.05). Values are means ± standard error (*n* = 3).

**Table 5 plants-12-01801-t005:** Effects of spraying KH_2_PO_4_ on grain quality parameters of wheat under heat stress during filling period.

Year	Treatment	PC (%)	TW (g·L^−1^)	WGC (%)	DT (min)	HAI (%)
2020–2021	CK	14.42 ± 0.25 a	799.33 ± 3.06 a	31.33 ± 0.40 a	4.17 ± 0.23 a	64.00 ± 1.00 a
	HT	13.69 ± 0.17 b	795.00 ± 4.36 a	29.92 ± 0.21 b	3.77 ± 0.06 b	62.33 ± 0.58 b
	KDP	14.38 ± 0.40 a	803.00 ± 2.00 a	31.31 ± 0.81 a	4.23 ± 0.12 a	64.33 ± 0.58 a
	PHT	13.75 ± 0.09 b	799.67 ± 0.58 a	30.11 ± 0.36 b	3.77 ± 0.06 b	63.00 ± 0.00 b
2021–2022	CK	15.23 ± 0.12 a	815.67 ± 2.08 a	33.07 ± 0.12 a	4.47 ± 0.12 a	72.00 ± 1.00 b
	HT	14.82 ± 0.02 b	814.00 ± 1.00 a	32.14 ± 0.06 b	4.07 ± 0.06 c	69.00 ± 1.00 c
	KDP	15.30 ± 0.04 a	818.00 ± 4.00 a	33.08 ± 0.15 a	4.53 ± 0.15 a	73.67 ± 0.58 a
	PHT	14.80 ± 0.03 b	816.67 ± 2.08 a	31.95 ± 0.06 b	4.27 ± 0.06 b	70.33 ± 0.58 c

CK, HT, KDP, and PHT represent spraying water, water combined with heat stress, 0.3% KH_2_PO_4_, and 0.3% KH_2_PO_4_ combined with heat stress, respectively. Different lowercase letters following the data in the same column indicate significant differences (*p* < 0.05). Values are means ± standard error (*n* = 3).

## Data Availability

All data supporting the results of this research are included within the article.

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
