# Peer review of "Effects of Spraying KH2PO4 on Flag Leaf Physiological Characteristics and Grain Yield and Quality under Heat Stress during the Filling Period in Winter Wheat"

_plants, 2023, doi:10.3390/plants12091801_

Round 1

Reviewer 1 Report

extremely well written, I have not suggestions for change.

Author Response

Comments and Suggestions for Authors

Extremely well written, I have not suggestions for change.

Response:

We are very appreciate your approval of our manuscript, which will be our motivation to continue our scientific work.

Reviewer 2 Report

The author described the objectives of article in a attractive way, and showed that article is of novel and in scope of journal. However, I have some minor suggestion before acceptance.

Comments: 

Please improve the quality of line 8-10.

Line 11: and quality of wheat.

Please revise the results part of Abstract is not in correct format. Please must show the value of important and significant variables. 

Also, mention the design of experiment.

Please arrange the keyword alphabetically.

Line 63: other environmental parameters (Akhtar et al., 2020).

 Akhtar, K.; Wang, W.; Ren, G.; Khan, A.; Nie, E.; Khan, A.; Feng, Y.; Yang, G.; Wang, H., Straw mulching with inorganic nitrogen fertilizer reduces soil CO2 and N2O emissions and improves wheat yield. Science of the Total Environment, 2020, 741, 140488.

Line 94-96: Optimizing water and fertilizer management is another critical aspect in controlling wheat growth, enhancing its resilience to adversity, and reducing the negative impacts of heat stress at harvest on yield (29,30, Zaheer et al., 2020).

Zaheer, S., Arif, M., Akhtar, K., …… Ain N-u. (2020). “Grazing and Cutting under Different Nitrogen Rates, Application Methods and Planting Density Strongly Influence Qualitative Traits and Yield of Canola Crop.” Agronomy 2020, Vol. 10, Page 404 10(3):404. doi: 10.3390/AGRONOMY10030404. 

Please highlights the importance and objective of your work in introduction section.

The start of your last paragraph of introduction is not good. Please avoid using words such as in conclusion in introduction.

You have presented your results very good but some places without values, please add it, such as in 3.1. Chlorophyll content in flag leaves, you have not added any values for this results. please check and fix this issue throughout the results section.

Discussion section is well written, however i suggest to add some recent citations of quality journal to improve the quality of your work. I have added some for your reference, please cite in discussion.

Haider, F. U., Wang, X., Farooq, M., Hussain, S., Cheema, S. A., Ain N-u et al. (2022). Biochar application for the remediation of trace metals in contaminated soils: Implications for stress tolerance and crop production. Ecotoxicol. Environ. Saf. 230, 113–165. doi: 10.1016/J.ECOENV.2022.113165. Ain N-u., Aslam A. and Haider, F. U. (2022). “Effects of Mulching on Soil Biota and Biological Indicators of Soil Quality” In “Mulching in Agroecosystems”, Eds. Kashif Akhtar, Muhammad Arif, Muhammad Riaz and Haiyan Wang, Springer Verlag, Singapore, 978-981-19-6409-1(ISBN)   

Author Response

Comments and Suggestions for Authors

The author described the objectives of article in a attractive way, and showed that article is of novel and in scope of journal. However, I have some minor suggestion before acceptance.

Comments: 

Please improve the quality of line 8-10. Line 11: and quality of wheat.

Response:

Thanks for your suggestion sincerely! According to your suggestion, we have revised this statement in the revised version in Page 1, Line 10-12.

Please revise the results part of Abstract is not in correct format. Please must show the value of important and significant variables. Also, mention the design of experiment. Please arrange the keyword alphabetically.

Response:

Thanks for your suggestion sincerely! According to your suggestion, we have revised the results part of Abstract and arranged the keyword alphabetically in the revised version in Page 1, Line 17-38, Line42-43.

Line 63: other environmental parameters (Akhtar et al., 2020).

Akhtar, K.; Wang, W.; Ren, G.; Khan, A.; Nie, E.; Khan, A.; Feng, Y.; Yang, G.; Wang, H., Straw mulching with inorganic nitrogen fertilizer reduces soil CO2 and N2O emissions and improves wheat yield. Science of the Total Environment, 2020, 741, 140488.

Line 94-96: Optimizing water and fertilizer management is another critical aspect in controlling wheat growth, enhancing its resilience to adversity, and reducing the negative impacts of heat stress at harvest on yield (29,30, Zaheer et al., 2020).

Zaheer, S., Arif, M., Akhtar, K., …… Ain N-u. (2020). “Grazing and Cutting under Different Nitrogen Rates, Application Methods and Planting Density Strongly Influence Qualitative Traits and Yield of Canola Crop.” Agronomy 2020, Vol. 10, Page 404 10(3):404. doi: 10.3390/AGRONOMY10030404.

Response:

Thanks for your suggestion sincerely! According to your suggestion, we have added the references in the Introduction section in Page 2, Line 86, Page 3, Line 125.

Please highlights the importance and objective of your work in introduction section. The start of your last paragraph of introduction is not good. Please avoid using words such as in conclusion in introduction.

Response:

Thanks for your suggestion! We have revised it in the revised version.

You have presented your results very good but some places without values, please add it, such as in 3.1. Chlorophyll content in flag leaves, you have not added any values for this results. please check and fix this issue throughout the results section.

Response:

Thanks for your suggestion! We have revised it in the revised version.

Discussion section is well written, however i suggest to add some recent citations of quality journal to improve the quality of your work. I have added some for your reference, please cite in discussion.

Haider, F. U., Wang, X., Farooq, M., Hussain, S., Cheema, S. A., Ain N-u et al. (2022). Biochar application for the remediation of trace metals in contaminated soils: Implications for stress tolerance and crop production. Ecotoxicol. Environ. Saf. 230, 113–165. doi: 10.1016/J.ECOENV.2022.113165. Ain N-u., Aslam A. and Haider, F. U. (2022). “Effects of Mulching on Soil Biota and Biological Indicators of Soil Quality” In “Mulching in Agroecosystems”, Eds. Kashif Akhtar, Muhammad Arif, Muhammad Riaz and Haiyan Wang, Springer Verlag, Singapore, 978-981-19-6409-1(ISBN)

Response:

Thanks for your suggestion! We have revised it in the revised version.

Reviewer 3 Report

Dears Authors,

In my opinion the Article entitled „Effects of spraying KH2PO4 on flag leaves physiological characteristics, grain yield and quality under heat stress during the filling period in winter wheat" can be published in the Scientific Journal - Plants, after minor revision. I enclosed the review in PDF file.

Corrections are connected with minor scientific errors and text editing. This needs to be supplemented or corrected. Enclosed please find the review of the above Article. Please carefully review all criticisms and suggestions.

Author Response

Comments and Suggestions for Authors

Dears Authors,

In my opinion the Article entitled „Effects of spraying KH2PO4 on flag leaves physiological characteristics, grain yield and quality under heat stress during the filling period in winter wheat" can be published in the Scientific Journal - Plants, after minor revision. I enclosed the review in PDF file.

Please give us the annual precipitation for the years of this study.

Response:

Thanks for your suggestion! We have added the annual precipitation during the two years of wheat growing seasons of this study in the revised version, please see Page 4, Line 158-160.

In my opinion, a brief characterization of the KH2PO4 (KDP) solution used is missing

Response:

Thanks for your comment! We have added the brief characterization of the KH2PO4 (KDP) solution used in the revised version, please see Page 4, Line 167-168.

Please explain this information? Was the experiment conducted as a field or in pots?

Response:

Thanks for your questions! This study was conducted under the field, and we have corrected the wrong word “pots” to “plots”. Please see Page 4, Line 188.

I would suggest that grain yield should be given in dt/ha or t/ha

Response:

Thanks for your questions! We have revised it in the revised version.

Corrections are connected with minor scientific errors and text editing. This needs to be supplemented or corrected. Enclosed please find the review of the above Article. Please carefully review all criticisms and suggestions.

Response:

Thank you very much for your comments and suggestions! We have revised it in the revised version.

Reviewer 4 Report

In relation to the manuscript, it represents an interesting study about fertilization at the foliar level and its effects against heat stress. However, there are aspects that need to be improved.

1. In the abstract, the abbreviations must be defined (SOD, CAT, POD, MDA) and quantitative information must be added in this results.

2. The introduction is mainly focused on providing information about photosynthesis, providing minimal information about antioxidant enzymes. This must be improved with updated information.

3. In the last paragraph of the introduction is there a conclusion? At the end of the introduction, a hypothesis and aims should be added.

4. In point 2.2, how were the times and doses for foliar fertilization defined?

5. How was basal fertilization defined?

6. In point 2.3.2. The same leaves were used for the photosynthetic and enzymatic measurements?

7. Point 2.3.5, please describe the methodologies.

8. In general terms, the description of the results is difficult to understand, with an excess of undefined abbreviations and writing problems. This section needs to be revised/rewritten. On the other hand, within the results, there are paragraphs that correspond to discussions (for example, lines 246-250,

9. Line 209, PDK or KDP?

10. The legends of figures 2, 3 and 4 must indicate in their legend or caption, all the used abbreviations.

11. In point 3.2, it is difficult to understand the description of the results when talking about Pn, since this abbreviation is not in the figure.

12. In table 1, express the results of DMM and DMR as tons per ha.

13. Lines 300-305, it is not understood. This paragraph needs to be rewritten. The same for lines 343-351 and 362-368.

14. The formats of tables 1, 2 and 5 should be improved, so that each result is only in one row.

15. Line 316, ANVOA?

16. Point 3.6, sucrose? Is that in table 5?

17. The discussion must be improved. It is very scarce and superficial, it necessarily requires improvement, including concrete, quantitative and updated information. In this section, there are several paragraphs that correspond to results and not to discussion.

18. Lines 400-403, the reference is missing.

19. The discussion regarding enzymatic activities needs to be improved.

20. The conclusions must be rewritten, according to the new version of the manuscript. Emphasizing the main aspects raised in the hypothesis and objectives

Author Response

Comments and Suggestions for Authors

In relation to the manuscript, it represents an interesting study about fertilization at the foliar level and its effects against heat stress. However, there are aspects that need to be improved.

  1. In the abstract, the abbreviations must be defined (SOD, CAT, POD, MDA) and quantitative information must be added in this results.

Response:

Thank you very much for your comments and suggestions! We have revised it in the revised version. Pleased see: Page 1, Line 17-38.

  1. The introduction is mainly focused on providing information about photosynthesis, providing minimal information about antioxidant enzymes. This must be improved with updated information.

Response:

Thank you very much for your suggestion! We have added more information about antioxidant enzymes in Introduction it in the revised version. Please see: Page 3, Line 100-103, Line 105-109.

  1. In the last paragraph of the introduction is there a conclusion? At the end of the introduction, a hypothesis and aims should be added.

Response:

Thank you very much for your question and suggestion! We have revised this inappropriate writing in the last paragraph of the introduction. And added a hypothesis and aims at the end of the introduction. Please see: Page 3, Line 134, Line 138-145.

  1. In point 2.2, how were the times and doses for foliar fertilization defined?

Response:

Thank you very much for your question! We have added reference this in Page 4, Line 176.

  1. How was basal fertilization defined?

Response:

Thank you very much for your question! The fertilizer application and schedule of wheat used in this study was recommended by the local wheat cultivation.

  1. In point 2.3.2. The same leaves were used for the photosynthetic and enzymatic measurements?

Response:

Thank you very much for your question! Before and after the heat stress of wheat, twenty flag leaves were randomly sampled from each experimental plot and divided into four equal parts to analyze the chlorophyll content (a+b), antioxidant enzyme activities and MDA content respectively. We have revised it in the revised version. Please see: Page 5, Line203.

  1. Point 2.3.5, please describe the methodologies.

Response:

Thank you very much for your suggestion! We have described the methodologies of grain quality parameters. Please see: Page 6, Line 237-241.

  1. In general terms, the description of the results is difficult to understand, with an excess of undefined abbreviations and writing problems. This section needs to be revised/rewritten. On the other hand, within the results, there are paragraphs that correspond to discussions (for example, lines 246-250,

Response:

Thank you very much for your comment and suggestion! We have revised it in the revised version. Please see: Page 6-13.

  1. Line 209, PDK or KDP?

Response:

Thank you very much for your question! We have revised it in the revised version. Please see: Page 6, Line 262.

  1. The legends of figures 2, 3 and 4 must indicate in their legend or caption, all the used abbreviations.

Response:

Thank you very much for your suggestion! We have added the specific information about the abbreviations in the revised version. Please see: Page 7-13.

  1. In point 3.2, it is difficult to understand the description of the results when talking about Pn, since this abbreviation is not in the figure.

Response:

Thank you very much for your comment! We have revised it in the revised version. Please see: Page 7, Line 291-294.

  1. In table 1, express the results of DMM and DMR as tons per ha.

Response:

Thank you very much for your comment! We have revised it in the revised version. Please see: Page 9, Line 347-349.

  1. Lines 300-305, it is not understood. This paragraph needs to be rewritten. The same for lines 343-351 and 362-368.

Response:

Thank you very much for your suggestion! We have rewritten this paragraph in the revised version. Please see: Page 10, Line 372-380; Page 12-13, Line 465-481.

  1. The formats of tables 1, 2 and 5 should be improved, so that each result is only in one row.

Response:

Thank you very much for your comment! We have revised it in the revised version.

  1. Line 316, ANVOA?

Response:

Thank you very much for your question! We have revised it in the revised version. Please see: Page 10, Line 403.

  1. Point 3.6, sucrose? Is that in table 5?

Response:

Thank you very much for your question! We are very sorry for the wrong information in the Figure 5 due to our mistake. We have revised it in the revised version. Please see: Page 12, Line 452-453.

  1. The discussion must be improved. It is very scarce and superficial, it necessarily requires improvement, including concrete, quantitative and updated information. In this section, there are several paragraphs that correspond to results and not to discussion.

Response:

Thank you very much for your comment and suggestion! We have improved the discussion in the revised version. Please see: Page 13-15.

  1. Lines 400-403, the reference is missing.

Response:

Thank you very much for your comment! We have added the reference in the revised version. Please see: Page 13, Line 526.

  1. The discussion regarding enzymatic activities needs to be improved.

Response:

Thank you very much for your comment and suggestion! We have revised it. Please see: Page 14, Line 531-534, 543-553.

  1. The conclusions must be rewritten, according to the new version of the manuscript. Emphasizing the main aspects raised in the hypothesis and objectives

Response:

Thank you very much for your comment! We have rewritten the conclusion in the revised version. Please see: Page 16, Line 636-650.

Round 2

Reviewer 4 Report

The manuscript was highly improved, now having a quality far superior to that of the first review. However, there are some aspects that still need to be improved:

1. Figures 3 and 4: the legend has an error in the format, they are divided above and below the figure.

2. The formats of tables 2 and 5 should be improved, so that each result is only in one row.

3. Although the discussion was improved, it was mainly with regard to explaining the results again. There is still a need to improve the discussion, including updated references.

Author Response

Comments and Suggestions for Authors:

The manuscript was highly improved, now having a quality far superior to that of the first review. However, there are some aspects that still need to be improved:

  1. Figures 3 and 4: the legend has an error in the format, they are divided above and below the figure.

Response:

Thanks for your suggestion sincerely! According to your suggestion, we have revised it in the revised version in Page 7 and 8.

  1. The formats of tables 2 and 5 should be improved, so that each result is only in one row.

Response:

Thanks for your suggestion sincerely! According to your suggestion, we have revised it in the revised version in Page 10 and 13.

  1. Although the discussion was improved, it was mainly with regard to explaining the results again. There is still a need to improve the discussion, including updated references.

Response:

Thanks for your comment and suggestion sincerely! Your professional comments are valuable and very helpful for improving our paper. We have improved the discussion according to your suggestion, please see: Page 13 -15.